# Influence of Interlayer Soil on the Water Infiltration Characteristics of Heavy Saline–Alkali Soil in Southern Xinjiang

Hongbo Liu [1,2,3], Bin Wu [1,*], Jianghui Zhang [2,3], Yungang Bai [2,3], Xianwen Li [4] and Bo Zhang [4]

[1]   College of Hydraulic and Civil Engineering, Xinjiang Agricultural University, Urumqi 830052, China; lhb090@163.com
[2]   Xinjiang Research Institute of Water Resources and Hydropower, Urumqi 830049, China; skyzjh@163.net (J.Z.); xjbaiyg@sina.com (Y.B.)
[3]   Key Laboratory of Saline-Alkali Soil Improvement and Utilization (Saline-Alkali Land in Arid and Semiarid Regions), Ministry of Agriculture and Rural Affairs, Urumqi 830052, China
[4]   College of Water Resources and Architectural Engineering, Northwest A&F University, Xianyang 712100, China; lixianwen@nwafu.edu.cn (X.L.); zhangbomg@nwafu.edu.cn (B.Z.)
*   Correspondence: wubin@xjau.edu.cn; Tel.: +86-991-8763365

**Abstract:** Interlayer soil is common in southern Xinjiang, because interlayer can reduce the infiltration rate of soil water. To simulate the interlayer soil in heavy saline–alkali cotton fields, this paper adopted a vertical one-dimensional infiltration test. T1 (315 mm), T2 (270 mm), and T3 (225 mm) and different interlayer positions (T5, 315 mm) and thicknesses of the interlayer (T6, 315 mm) with the same irrigation volume, as well as one perforation and sand filling treatment (T4, 315 mm), were set. The influence of different irrigation amounts, locations, and thicknesses of the interlayer and sand injection on water infiltration was analyzed. The analysis results showed that with the increase in irrigation amount, the water infiltration rate and the migration distance of the wet front increased, but did not penetrate to the bottom soil (90 cm). Under the same irrigation volume, the increase in interlayer thickness (T6) compared with the increase in interlayer position (T5), the change in soil moisture content in the upper and lower layers of the interlayer was greater, and the advance time of wetting front migration and cumulative infiltration were slightly higher. After tunneling and sand filling (T4), the infiltration rate of water was increased, the migration time of the wet front was reduced, and the profile water content of each soil layer was improved. The Kostiakov model could better simulate the water infiltration characteristics of interlayer soil with different profile configurations in heavily saline–alkali land. The results showed that in all of the treatments, only the wet front of the soil moisture reached 100 cm in the T4 treatment, and the maximum was only 87.8 cm in the other treatments, indicating that too little irrigation water or the upward movement and thickening of the interlayer were not conducive to water infiltration. For the interlayer soil area in the heavy saline–alkali land in southern Xinjiang, the appropriate irrigation water should be more than 315 mm. The treatment of drilling first and then filling sand can be used as a simple but effective measure to increase the water infiltration rate of the interlayer soil, and can thus be applied to the layered soil structure in the interlayer position of 60–80 cm.

**Keywords:** southern Xinjiang region; soil column test; drilling and sand filling; interlayer soil; water infiltration characteristics

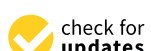



## 1. Introduction

The total area of saline and alkaline land in China is 9913 × 10⁴ ha; less than 20% of the total area of saline and alkaline land has been cultivated and approximately 80% of saline and alkaline land has the potential to develop into cultivated land [1]. Among the 1031 × 10⁴ ha of future usable wasteland in Xinjiang, the area of salinization is 515.11 × 10⁴ ha,

accounting for 49.93% of the total area [2]. Among the total arable land in Xinjiang, the area containing interlayers is $32.5 \times 10^4$ ha, accounting for approximately 8.0% of the total arable land area. In areas with more than moderate salinization, laminated soil is more common, which seriously restricts the sustainable development of irrigation agriculture in southern Xinjiang.

Soil profiles in the northwestern China are usually layered, which often include different interlayer patterns, such as sandy soil with a clay layer in the middle, or sandy soil with a sand layer in the middle, or sand and clay alternatively interlaid. The water holding capacity and infiltration of layered soil are very different from those of homogeneous soil. Under the condition of a layered structure, the heterogeneity of soil texture causes sudden changes in soil–water potential at the interface, resulting in changes in the movement mode of infiltration water at the interface [3–5]. For example, Si et al. [6] found that the discontinuity of pores at the interface of stratified soil affected the water distribution and water flux in the whole soil body. In early studies on the infiltration rules of layered soil, Colman et al. [7] proposed that each layer of soil could be regarded as homogeneous soil, and the infiltration process was controlled by fine soil, whether fine soil covered coarse soil or coarse soil covered fine soil. Miller et al. [8] found that regardless of the order of thickness and fineness of two layers of soil, as long as the texture and water conductivity of the subsoil were different from those of the surface soil, the effect would be to reduce the infiltration rate. When there is an interlayer in the soil profile, no whether it is matter clay over sand or sand over clay, it will inhibit water infiltration [9], but if the soil is alternatively interlaid with clay and sand layers, the mixed structure will increase the overall water holding capacity of the profile [10,11]. Meanwhile, the interlayer structure not only changes the soil water infiltration performance, but it also affects the water distribution [12]. For example, different irrigation amounts [13], textures of interlayer soil [14,15], and irrigation methods [16,17] will all have an impact on water infiltration.

Although relatively rich research achievements have been made in the study of the water infiltration regularity of interlayer soil, studies on the water movement of interlayer soil in the heavily saline–alkali land in southern Xinjiang have not been reported. In this regard, in view of the fact that interlayer soil is common in heavy saline–alkali soil after the construction of high-standard farmland and large-scale water-saving transformation in southern Xinjiang, this paper studies the mechanism of water infiltration and movement in this heavy saline–alkali interlayer soil and analyzes the influence of different irrigation amounts, interlayer locations, and interlayer thicknesses on the soil infiltration rule through a vertical one-dimensional infiltration test. The study of interlayer soil water infiltration characteristics under different scenarios provides theoretical and data support for the rational development and utilization of heavy saline–alkali land in southern Xinjiang.

## 2. Materials and Methods

### 2.1. Test Conditions

The experimental area is located in the 3333 ha high-efficiency water-saving and income-increasing pilot area of Shaya (41°14′530″ N, 82°43′33″ E). The soil was collected separately according to the original soil layer and transported back to the laboratory. The soil was successively air-dried, rolled, ground, and screened by 2 mm on the drying board and then mixed evenly according to the soil layer for use. The mechanical composition of the test soil was measured using an EyeTech laser particle size analyzer. The soil layers were silty loam. Afterbeing sampled with cutting rings, the soil bulk density was calculated from the dry soil mass over the ring volume, and the field capacity was obtained by dividing the weight difference between the saturated and the air-dried soil over the air-dried soil mass. In order to minimize the undesirable variations in soil moisture, all of the soil was air dried to obtain a comparatively consistent water content. Therefore, the soil moisture after air-drying was recorded as the initial soil moisture using TDR (Kaimeite, Zhuhai, China). The determination of soil physical parameters was entrusted to the Xinjiang Institute of Ecology and Geography.

## 2.2. Soil Layer Configuration Design

To fully reflect the spatial difference of interlayer soil in the field, six treatments were set up for the test soil column, among which three were treated with different irrigation quotas, with irrigation quotas of 315 mm (T1), 270 mm (T2), and 225 mm (T3). The results will be used in conjunction with the actual irrigation quota of the field in winter. The thickness of the interlayer was 10 cm. They were all located at 60–70 cm. The interlayer thickness of 10 cm was 225 mm, and the interlayer position was 60–70 cm. A plastic pipe with holes was inserted in the center, the diameter of the holes was 3 mm, the spacing of the holes was 1 cm, the inner diameter of the pipe was 2 cm, the outer diameter of the pipe was 2.6 cm, the insertion depth was 80 cm, and the pipe was filled with quartz sand (2 mm diameter). The irrigation quota for the interlayer location test treatment (T5) was 315 mm, the interlayer thickness was 10 cm, and the location was 50–60 cm. The irrigation quota for the interlayer thickness test treatment (T6) was 315 mm, the interlayer thickness was 15 cm, and the location was between 60 cm and 75 cm. In the experiment, the T1–T4 treatments were filled according to the capacity of each soil layer in Table 1, while the T5 treatments were filled with interlayer soil at 50–60 cm, with a capacity of 1.73 g·cm$^{-3}$. The original 50–60 cm soil layer was filled at 60–70 cm, while other layered soils remained unchanged. The T6 treatment was used to fill the interlayer soil at a depth of 60–75 cm, and 75–100 cm was filled in the original layers.

**Table 1.** Basic physical properties of the tested soil.

| Soil Depth (cm) | Particle Composition (%) | | | Texture | Bulk Density (g·cm$^{-3}$) | Initial Moisture Content (cm$^3$·cm$^{-3}$) | Field Capacity (cm$^3$·cm$^{-3}$) | Salt Content (g·kg$^{-1}$) | pH |
|---|---|---|---|---|---|---|---|---|---|
| | <0.002 mm | 0.002–0.02 mm | 0.02–2 mm | | | | | | |
| 0–10 | 5.87 | 65.50 | 28.63 | Silty clay | 1.65 | 0.056 | 0.352 | 14.09 | 8.71 |
| 10–20 | 5.51 | 65.38 | 29.11 | Silty clay | 1.56 | 0.065 | 0.337 | 11.68 | 8.70 |
| 20–30 | 6.16 | 69.40 | 24.44 | Silty clay | 1.51 | 0.073 | 0.330 | 10.79 | 8.70 |
| 30–40 | 5.88 | 68.07 | 26.05 | Silty clay | 1.61 | 0.071 | 0.400 | 10.86 | 8.69 |
| 40–50 | 4.32 | 65.82 | 29.86 | Silty clay | 1.38 | 0.063 | 0.329 | 21.18 | 8.74 |
| 50–60 | 5.77 | 74.03 | 20.20 | Silty clay | 1.56 | 0.071 | 0.312 | 24.07 | 8.67 |
| 60–70 | 6.75 | 72.65 | 20.60 | Silty clay | 1.73 | 0.069 | 0.284 | 29.97 | 8.79 |
| 70–80 | 4.68 | 62.12 | 33.20 | Silty clay | 1.69 | 0.067 | 0.398 | 24.06 | 8.78 |
| 80–90 | 4.29 | 58.16 | 37.55 | Silty clay | 1.55 | 0.063 | 0.410 | 19.77 | 8.77 |
| 90–100 | 5.41 | 71.00 | 23.59 | Silty clay | 1.57 | 0.064 | 0.381 | 16.12 | 8.81 |

## 2.3. Test Apparatus and Procedure

The test device is shown in Figure 1. The inner diameter of the plexiglass tube was 20 cm, the height was 140 cm, the bottom of the soil column was filled with 5 cm quartz sand, the 0–100 cm layer was strictly packed in layers according to the original soil, and the corresponding quality was weighed according to each layer at intervals of 5 cm. To avoid the stratification of the soil column in the test process, it is necessary to scratch the soil surface and then fill the next layer. For each treatment, the DJS-10C soil three-ginseng sensor probe was inserted at 10, 30, 50, 70, and 90 cm, and the soil parameter acquisition system (BSI-DS16T) was connected. After filling the soil column, the soil surface was covered with filter paper and allowed to stand for 48 h. After the infiltration test began, water of a pre-designed volume (i.e., simulated irrigation height) was fed from the top of each column all at one time to start the infiltration test. The water heights of T1, T2, and T3 were 315 mm, 270 mm, and 225 mm, respectively, and those of T4, T5, and T6 were all 315 mm. The position of the wetting front was recorded, and the cumulative infiltration amount was read according to the time density first and then thinning. Meanwhile, the step size of the data collector was set to automatically record the changes in the soil moisture content, salt content, and temperature at the sensor probe in the infiltration process with time every 5 min. After all of the water had penetrated, the wetting front did not change significantly, and the infiltration test was over.

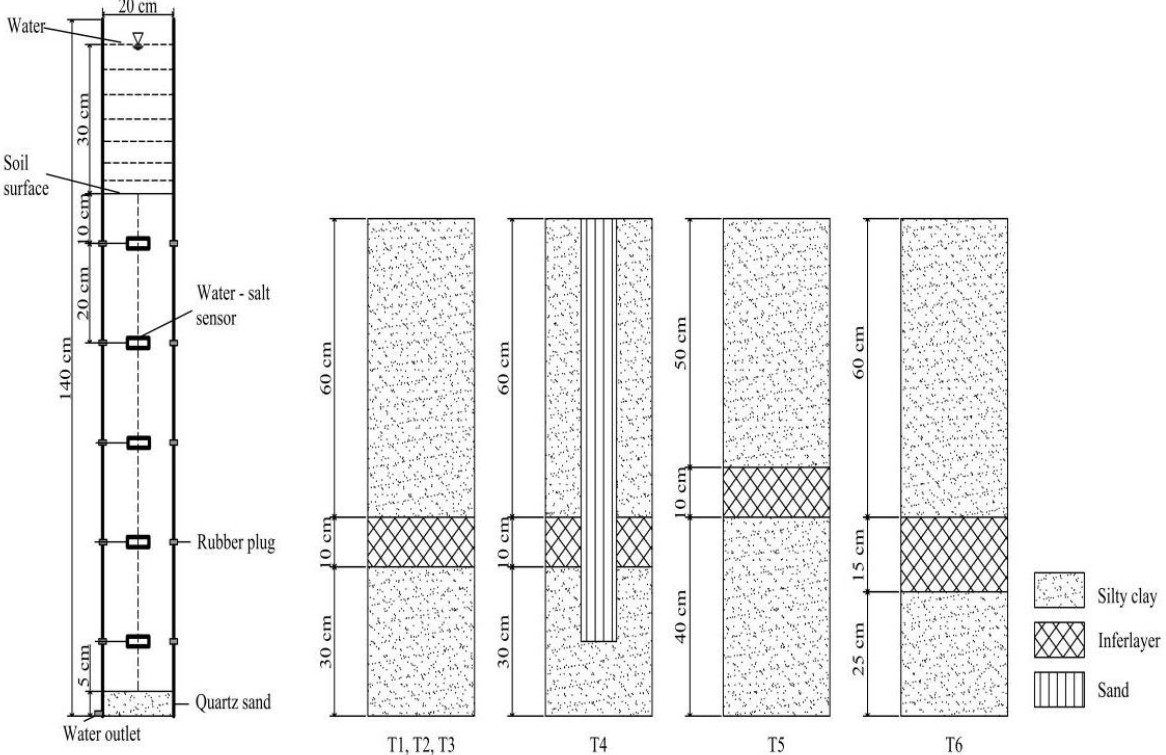

**Figure 1.** Test soil column and sensor position and interlayer profile structure diagram.

*2.4. Infiltration Model and Evaluation Index*

Kostiakov is a common empirical model that is easy to calculate and can better reflect the infiltration characteristics of soil. In this paper, the least squares method was used to fit the experimental data of different treatments, and the applicability of the model to simulate the soil water infiltration process in the interlayer soil profile (T1–T6) was analyzed.

The Kostiakov infiltration model is as follows:

$$I(t) = kt^n (k > 0, 0 < a < 1) \tag{1}$$

where I(t) is the cumulative infiltration amount, cm; T is the infiltration time, min; and k and n are empirical parameters of the model. The determination coefficient ($R^2$) and relative root mean square error (RRMSE) are used as index parameters to evaluate the simulation effect of soil water infiltration using the Kostiakov model. The closer the $R^2$ value is to 1, the smaller the RRMSE value is, indicating a better fitting effect of the selected infiltration model.

*2.5. Data Analysis*

WPS2022 (Zhuhai Jinshan Office Software Co., Ltd., Zhuhai, China) was used for data processing and analysis, Origin2021 (MicroCal, Northampton, MA, USA) software was used for charting, and SPSS26.0 (IBM, Amonk, New York, NY, USA) software was used for correlation and regression analysis.

**3. Results**

*3.1. Effect of Interlayer Soil on Infiltration Rate*

The infiltration rate is influenced by many factors, such as the initial soil moisture content and texture, soil structure, and water supply. The specific physical parameters of the soil are shown in Table 1.

The experiment first studied the influence of different irrigation amounts on the infiltration rate under the condition that the physical characteristics of each soil layer

remained unchanged and then determined the ideal irrigation amount. Then, under the condition of a constant irrigation amount, the influence of the change in sand, interlayer location, and interlayer thickness on the infiltration rate was studied. To better explain the influence of interlayer soil on the soil infiltration rate, the overall stable infiltration rate of the soil, the average infiltration rate of the soil layer above the interlayer, and the average infiltration rate of the interlayer were used for the calculation and analysis.

The overall infiltration rate of soil columns in each treatment is shown in Figure 2. As the infiltration rate represents the change in infiltration depth with time, the 100 cm soil layer was divided into 10 points according to the infiltration depth of each treatment in the soil column test. Therefore, the corresponding values of the depth of each treatment (experimental points) were between 7–10 points. The changes for each treatment were consistent over time, showing a trend of first rapid decrease and then gradually becoming stable over time. The main reason is that in the early stage of infiltration, the initial moisture content of the soil surface was very low. Combined with the adsorption of soil capillary force, soil has high soil water suction at the initial stage of infiltration and the corresponding infiltration rate is very high [18]. Over time, soil water suction decreases with the increasing water content, and the infiltration rate also tends to be stable. There were some differences in the stable infiltration rate of each treatment. Generally, T1, T2, T3, and T4 tended to be stable after 1500 min, and T5 and T6 tended to be stable after 5000 min.

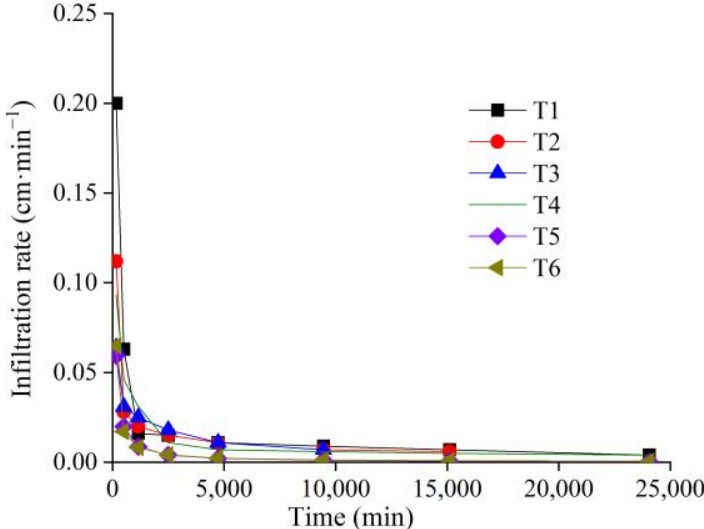

**Figure 2.** Change in the infiltration rate of each treatment with time.

The stable infiltration rate was the ratio of the cumulative infiltration amount after the stabilization of infiltration to the infiltration time. As the interlayer divides the soil into "upper layer" and "lower layer", the water infiltration characteristics of different interlayer soils were different. As an important parameter of the water infiltration characteristics of interlayer soil, the stable infiltration rate reflects the infiltration characteristics of the overall structure of interlayer soil, as shown in Table 2. The irrigation amounts of T1, T2, and T3 were different, and the stable infiltration rate showed a decreasing trend with the decreasing irrigation amount, but the difference was small. The stable infiltration rates of T1, T2, and T3 were 0.226 cm·h$^{-1}$, 0.225 cm·h$^{-1}$, and 0.213 cm·h$^{-1}$, respectively. The T4 treatment had the highest stable infiltration rate of 3.509 cm·h$^{-1}$ because irrigation water could quickly reach the bottom of the soil after drilling holes and filling sand in the middle of the soil column, which increased the infiltration rate by 15.5 times compared with T1. When the interlayer thickness of the T6 treatment increased from 10 cm to 15 cm, the stable infiltration rate decreased by 55.64% compared with that of T1. After the interlayer position of the T5 treatment was moved from 60–70 cm to 50–60 cm, the stable infiltration rate further decreased, which was significantly lower than that of the other treatments and was

0.066 cm·h$^{-1}$, which was only 29.17% of that of the T1 treatment. The results show that the infiltration effect can be significantly increased by drill holes and sand filling, and the permeability reduction effect can be significantly improved by increasing the thickness of the interlayer and moving up the position of the interlayer.

**Table 2.** Statistics of the infiltration characteristics of each treatment.

| Treatment | Distance of Clay Interlayer from Soil Surface/cm | Stable Infiltration Rate/(cm·h$^{-1}$) | Mean Infiltration Rate of Sediment Layer above Clay Interlayer/(cm·h$^{-1}$) | Mean Infiltration Rate of Clay Interlayer/(cm·h$^{-1}$) |
|---|---|---|---|---|
| T1 | 60 | 0.226 | 3.696 | 0.570 |
| T2 | 60 | 0.226 | 2.238 | 0.426 |
| T3 | 60 | 0.213 | 1.566 | 0.420 |
| T4 | 60 | 3.509 | 2.250 | 0.348 |
| T5 | 50 | 0.066 | 1.380 | 0.126 |
| T6 | 60 | 0.100 | 1.164 | 0.078 |

In terms of the average infiltration rate from the wetting front to the penetration of the soil layer above the interlayer, the change law of the average infiltration rate of each treatment was similar to that of the stable infiltration rate. Compared with the T2 and T3 treatments, the T1 treatment sped up the infiltration of water under the action of gravity due to the maximum irrigation volume. The average infiltration rate of the T1 treatment was 39.45% and 57.63% higher than that of T2 and T3, respectively. Compared with the other treatments, the irrigation water in the T4 treatment penetrated deeper into the soil faster and then started to penetrate. The average infiltration rate of the T4 treatment was lower only than that of the T1 treatment, with an average infiltration rate of 2.25 cm·h$^{-1}$. The average infiltration rate of the soil layer above the interlayer treated with T5 was only higher than that treated with T6, at 1.38 cm·h$^{-1}$, while that of the soil layer treated with T6 was the lowest, at 1.164 cm·h$^{-1}$. This is because after the interlayer treated with T5 moved upward, the soil layer above the interlayer changed from 60 cm to 50 cm, but because the overall stable infiltration rate was the lowest, the average infiltration rate to the soil layer above the interlayer was lower. Compared with the other treatments, the T6 treatment had the lowest infiltration rate, which was affected by the overall stable infiltration rate and the thickness of the soil layer above (60 cm). When the wetting front reached the interlayer, due to the water conduction barrier, the infiltration rate was further reduced. The average infiltration rate of interlayers with different treatments is shown as follows: T1 > T2 > T3 > T4 > T5 > T6. Among these, the infiltration rate of the interlayer soil treated by T1 was the highest, at 0.57 cm·h$^{-1}$. This is because the infiltration rate of the interlayer soil treated by T1 was accelerated under the action of gravity under the condition of the same irrigation volume. In contrast, the irrigation volume decreases, and the gravity effect of water also decreases accordingly. Compared with interlayer moving up (T5) and thickening (T6), the average infiltration rate of the interlayer was increased by 63.79% and 77.59, respectively, in the T4 treatment, which also indicates that the interlayer moving up and thickening increased the permeability reduction effect of the interlayer.

The above analysis shows that the permeability reduction effect was different with different irrigation amounts, interlayer locations, and interlayer thicknesses. With the thickening of the interlayer, the permeability reduction effect showed an increasing trend. When the interlayer thickness was 15 cm, the permeability reduction effect was the strongest, followed by when the interlayer was 50 cm away from the topsoil. The experimental data show that the overall infiltration rate of the soil layer was significantly improved, the permeability reduction effect of the interlayer was improved, and the water infiltration was promoted.

*3.2. Influence of Interlayer Soil on Wetting Front Migration*

This can be seen from the change in the migration distance of the wetting front with time in each treatment (Figure 3). Under different irrigation conditions, comparing the three treatments, T1, T1, and T3, it can be seen that, influenced by the amount of irrigation, there was a significant difference in the movement distance of the wetting front in each treatment. The movement distance of the wetting front increased with increasing the irrigation water. Because the T1 treatment had the largest amount of irrigation, the advancing distance of the wetting front in the T1 treatment also increased, reaching 87.8 cm. Compared with T1, the irrigation amount in the T2 and T3 treatments decreases by 14.29% and 28.57%, respectively, and the advancing distance of the wetting front decreased by 17.31% and 21.18%, respectively. Moreover, T2 and T3 were affected by the amount of irrigation, and their maximum advancing distances from the wetting front were 72.6 cm and 69.2 cm, indicating that the amount of irrigation treated by T3 was not sufficient to penetrate the interlayer. For the same irrigation amount, comparing the T1, T4, T5, and T6 treatments, it was found that the T4 treatment accelerated water infiltration by penetrating the interlayer through drilling and sand filling, and the wetting front advancement rate was significantly higher than that of the other treatments. Not only did a high wetting front advancement rate significantly improve the total time from the beginning of infiltration to the end of infiltration, which was only 495 min, but the wetting front advancement distance also exceeded 100 cm, resulting in leakage. Because of the upward displacement of the interlayer by 10 cm and the thickening of the interlayer by 5 cm, the movement of the wetting front in the T5 and T6 treatments was consistent, but the advancing distance of the T5 treatment was 81.7 cm, which was lower than that of the T6 treatment (88.0 cm). Moreover, the advancing time of the wetting front in treatment T2 was significantly slower than that in treatment T1, with the total maximum advancing distances being 20,050 min and 13,260 min, which were 3.5 and 2.3 times as long as that in treatment T1, respectively.

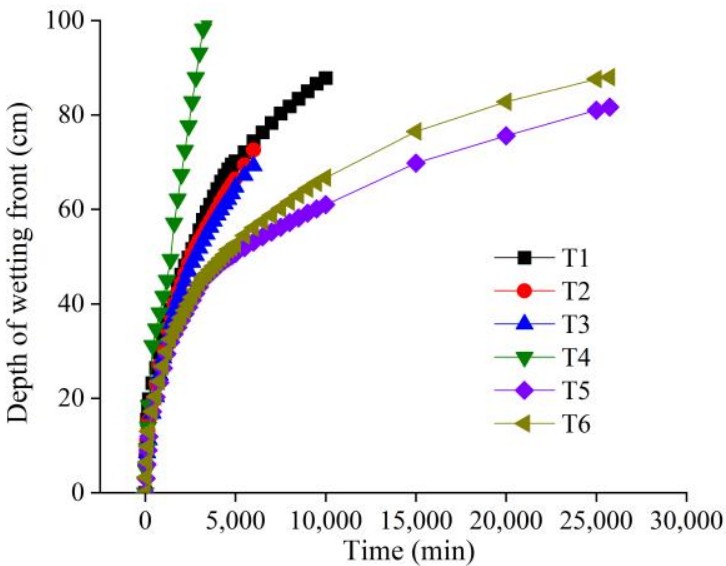

**Figure 3.** Change in wetting front with time in each treatment.

In general, all treatments showed roughly the same change rule in the depth for humid front migration. The T4 treatment was significantly higher than the other treatments in the migration rate of the humid front because it perforated the interlayer at the central point. Because of the infiltration reduction effect of the interlayer, the amount of irrigation had a significant impact on the wetting front. If the amount of irrigation was too small, the entire interlayer (T3) could not be wetted. Moreover, using the field-planned irrigation amount treatment (T2), the wetting front depth was only 72.6 cm, only passing through the interlayer. This indicates that in the presence of an interlayer in the field, to make the

soil moist to the soil layer below the interlayer, the irrigation amount needed to be above 270 mm. At the same time, due to the different location and thickness of the interlayer, the time of the wetting front passing through the soil above the interlayer into and through the interlayer was also different, and the average infiltration rate of the T5 and T6 treatments above the interlayer and the average infiltration rate of the interlayer were lower than those of the other treatments. Therefore, the advancing duration of the wetting front was significantly higher than that of the other treatments, and the upward movement and thickening of the interlayer could significantly reduce the soil water infiltration; that is, the upward movement or thickening of the interlayer could enhance the permeability reduction effect. The T4 treatment accelerated water infiltration to a certain extent, not only improving the overall stability rate of each soil layer, but also accelerating the average infiltration rate of the interlayer and the infiltration rate above the interlayer. However, after irrigation, water infiltration was too fast, and if the irrigation amount was too large, deep leakage formed, resulting in a waste of water resources.

*3.3. Effect of Interlayer Soil on Cumulative Infiltration*

The cumulative infiltration amount refers to the total amount of water that seeps into the soil per unit area of the surface within a certain period of time after the beginning of infiltration. The cumulative infiltration amount for each treatment varied over time, as shown in Figure 4. In the early stage of infiltration, the gradient of water potential was large, and the cumulative infiltration increased rapidly, especially in the T4 treatment, because the 0–80 cm depth of the soil column in the T4 treatment was homogeneous sand with good permeability, so the infiltration was fast. When the soil reached the bottom layer, the water penetrated to the periphery at the same time, so the cumulative infiltration shows a slowing trend after 200 min of infiltration. However, its infiltration rate was still higher than that of the other treatments, and the cumulative time was only 495 min. In the other treatments, the cumulative infiltration amount varied with time in a consistent manner and presented a nonlinear change; that is, before reaching the interlayer interface, the slope of the cumulative infiltration amount was larger, and the cumulative infiltration amount varied linearly with time. When passing through the interlayer, the slope decreased. When the infiltration front passed through the interface between the interlayer and the lower layer, the slope of the cumulative infiltration amount curve increased again, but the degree of increase was weaker than the slope before reaching the interlayer. As a result of the fixed amount of irrigation water, the infiltration test was completed when the fixed amount of water completely penetrated the surface of the soil column, so the cumulative infiltration amount for the T1, T4, T5, and T6 treatments was 31.5 cm; that for T2 was 275 cm; and that for T3 was 22.55 cm. However, influenced by factors such as irrigation amount and interlayer location, there were significant differences in infiltration time among the treatments, with infiltration times of 5730 min for the T1 treatment and 4885 min and 4100 min for the T2 and T3 treatments, respectively, indicating that the infiltration time decreased with the decreasing irrigation amount when the structure of each soil layer remained unchanged. However, when the interlayer was moved up and thickened, the infiltration time increased significantly, such as 20,050 min for the T5 treatment and 13,260 min for the T6 treatment. This indicates that under the condition of a certain amount of irrigation, the upward movement or thickening of the interlayer significantly enhanced the permeability reduction effect and greatly prolonged the infiltration time, thus restricting water infiltration to a certain extent. However, after penetrating the interlayer and sand filling, the original water migration law was be changed due to the distribution of pores in the reconstructed soil profile, the size of the water crossing section, and the movement channel of soil water. At the same time, the existence of large pores accelerated the water infiltration and shortened the time of water infiltration greatly.

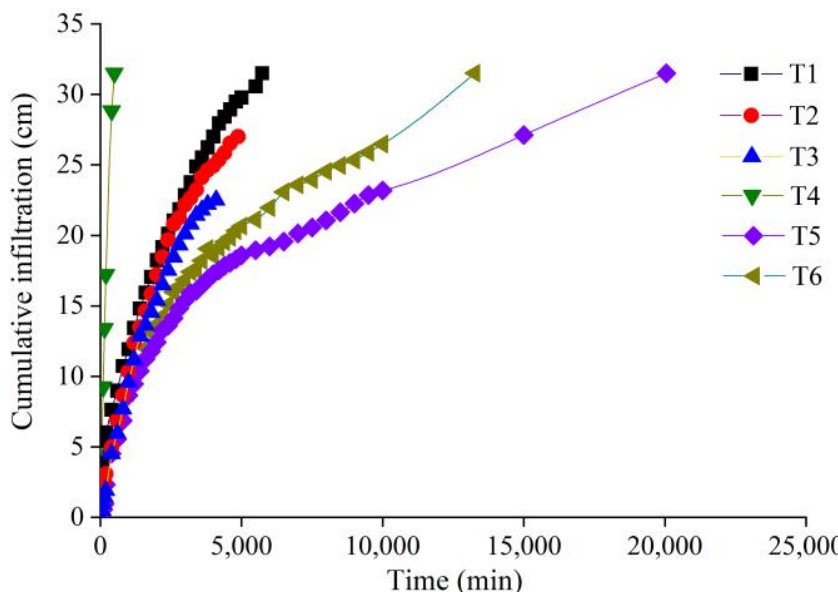

**Figure 4.** Change in cumulative infiltration with time for each treatment.

### 3.4. Influence of Interlayer Soil on the Relationship between Cumulative Infiltration and Wetting Front

By comparing the relationship between the cumulative infiltration (I) and wetting front migration distance (Zf) of different treatments in the infiltration process, as shown in Table 3, it was found that there was a significant linear relationship between the two ($R^2$ > 0.992), and the coefficient of a had a linear slope. This was expressed as the amount of water required to advance a wetting front per unit distance, and can also reflect the water holding capacity of the soil profile. As a result of different irrigation volumes and the relationship between the position and thickness of the interlayer, the variation range of the a value was different, which also reflects the difference in water holding capacity between different treatments. For example, for the T4 treatment, due to the small difference between the cumulative infiltration amount and the advancing distance of the wetting front at the same time, the a value was larger, and for the T5 treatment, due to the effect of the interlayer moving upward to prevent the water flow from continuing to flow downward, the infiltration time was relatively longer, and the water holding capacity was relatively low before the amount of infiltration reached its maximum water holding capacity. Therefore, the relationship curve between cumulative infiltration and wetting front also reflects the overall bearing capacity of soil, providing a certain reference basis for formulating a reasonable irrigation system.

**Table 3.** Linear regression parameters of cumulative infiltration (I) with wetting front migration distance (Zf) in each treatment.

| Treatments | Fitting Equation (I = f(Zf)) | $R^2$ |
|:---:|:---:|:---:|
| T1 | I = 0.4603 Zf − 2.7518 | 0.9963 |
| T2 | I = 0.4651 Zf − 3.2838 | 0.9923 |
| T3 | I = 0.4243 Zf − 2.3322 | 0.9971 |
| T4 | I = 0.8993 Zf + 0.6224 | 0.9991 |
| T5 | I = 0.4118 Zf − 2.3074 | 0.9921 |
| T6 | I = 0.4298 Zf − 2.1399 | 0.9956 |

### 3.5. Variation in Water Content in Different Treatment Profiles

The changes in soil profile moisture content over time at different soil depths in each treatment are shown in Figure 5. During the infiltration process, by comparing the soil



drying method with the profile water content of the automatic data recording and collection system at the same time, it was found that some probes measured a slightly higher profile water content, but it did not affect the change trend of soil water in the whole infiltration process [19]. With the downward advance of the wetting front, the moisture content of different soil layers under different treatments fluctuated but showed a certain pattern, and the soil moisture content of each soil layer increased sharply and tended to be stable. As a result of the different treatments, the time when the soil moisture content of each soil layer became stable showed obvious differences; for example, in the T4 treatment, the soil moisture content of each layer gradually became stable on the fifth day after the beginning of infiltration, and the stabilization time was earlier than that of the other treatments.

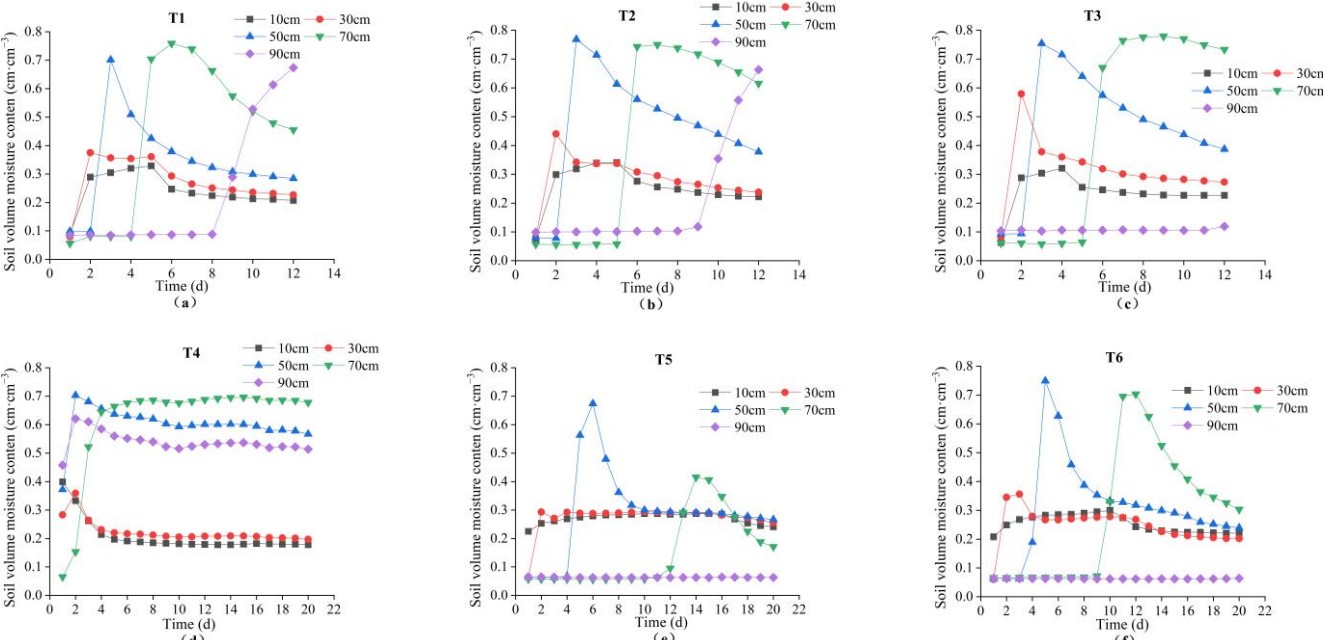

**Figure 5.** Change in the water content of the soil profile with time in each treatment. (**a**,**b**,**c**,**d**,**e** and **f**) respectively represent T1–T6 processing.

By comparing the variation trend of the water content of the profile treated by T1–T3 over time, it was found that although the structure of each soil layer was the same in each treatment, due to the existence of the interlayer and the difference in irrigation amount, the water content of the interlayer soil increased first and then decreased. However, with the decrease in irrigation amount, the water content increased with the time lag; for example, the T1 treatment reached the maximum value on the fifth day. The T2 and T3 treatments reached the maximum value on the sixth and eighth days, respectively, and the decreasing trend slowed down with the decrease in irrigation amount, except for the depth of 90 cm in the other soil layers. The change trend was consistent with that of the interlayer. Because of the difference in the amount of irrigation, the smaller the amount of irrigation, the smaller the change in the deepest soil layer (90 cm); among them, the wetting front of the T5 and T6 treatments at 25,000 min after the start of the experiment was 81.0 cm and 87.6 cm, respectively. Therefore, at a depth of 90 cm, the soil moisture content remained in a stable state of "no change" within 20 days after the start of irrigation. At the same time, it can be seen that after the interlayer soil was broken by drilling, the distribution of moisture content in the upper and lower soil profiles was significantly improved. For example, after the T4 treatment, the soil moisture content in each soil layer increased significantly after irrigation, and with the extension of time after irrigation, the moisture content in each soil layer decreased and tended to be stable after the fourth day. Under the same irrigation quantity, the change law of the soil moisture content in the case of the interlayer moving up

was consistent with that in the case of the thickness increasing, but the peak time of the soil moisture content in the interlayer moving up was later than that in the case of thickening.

Through the above analysis, it was found that under different irrigation conditions, the greater the irrigation amount (315 mm), the faster the water content of the surface soil increased. As the irrigation continued, the water content of the 50 cm soil layer began to increase. When the irrigation water seeped into the interlayer, the water content of the interlayer soil began to increase, and while the water content of the upper layer rapidly decreased, the water content of the interlayer soil rapidly decreased after reaching the peak value; after 8 days of irrigation, the soil moisture content of the lowest layer (90 cm) began to increase. In contrast, when the amount of irrigation water (270 mm) decreased, although the trend of soil moisture content change at each depth at the initial stage of irrigation was consistent with that at 315 mm, affected by the amount of irrigation, the increase in soil moisture content from the interlayer and below was delayed. Similarly, when the irrigation volume was 225 mm, the change trend of the soil moisture content at different depths was consistent 5 days after irrigation, but after that, the change in the soil moisture content at the interlayer (70 cm) and bottom layer (90 cm) was obvious. The interlayer soil began to decrease after reaching the peak value on the sixth day, but the decrease range was small; however, the bottom layer did not increase until 12 days after irrigation. As a result of the changes in irrigation amount, thickness, and position, the time of water infiltration to the next layer also changed. Therefore, the moisture content at the same soil depth at the same time was different, but it still showed similar patterns after a certain period of time; for example, when the irrigation amount was different and the thickness of the interlayer was the same as that of the position (T1, T2, and T3), or the position and thickness of the interlayer were different and the irrigation volume was the same (T5 and T6). This shows that the interlayer had little influence on the moisture content of the surface soil, but has a greater influence on the moisture content of the upper and lower layers of the interlayer; that is, it was difficult for water to penetrate through the interlayer to the lower layer, and the increase in the interlayer thickness had a greater influence on the moisture content of the upper and lower layers of the interlayer than the upward movement of the interlayer.

### 3.6. Simulation Analysis

The Kostiakov model was applied to fit the relationship between the measured infiltration amount I (cm) and the infiltration time t (min) of each treatment soil, and the simulation results are shown in Table 4. The $R^2$ values fitted by the Kostiakov model for each treatment were all above 0.93, and the RRMSE was no more than 0.25, except for the T5 treatment (0.8986), indicating that the Kostiakov model could simulate the infiltration characteristics of soil water in interlayers with different profile configurations in saline–alkali soil well. Therefore, the data are not logarithmically transformed into a linearized form of the equation [20].

**Table 4.** Infiltration simulation by the Kostiakov model for different treatments.

| Treatment | Model of KostiakovI(t) = kt$^n$ | | $R^2$ | RRMSE |
|:---:|:---:|:---:|:---:|:---:|
| | **k** | **n** | | |
| T1 | 0.2642 | 0.5567 | 0.9829 | 0.071 |
| T2 | 0.0327 | 0.8142 | 0.9728 | 0.103 |
| T3 | 0.0191 | 0.8759 | 0.9873 | 0.121 |
| T4 | 0.2928 | 0.7525 | 0.9830 | 0.246 |
| T5 | 0.0333 | 0.7440 | 0.8986 | 0.077 |
| T6 | 0.0554 | 0.6985 | 0.9399 | 0.077 |

## 4. Discussion

### 4.1. Influence of Interlayer Soil on Moisture Content

In the field, the soil profile is rarely uniformly distributed, and the overall superstructure of the soil can be divided into an interlayer and an overlying layer. The interlayer

refers to a layer of clay or sand in a homogeneous soil mass, and the overlying layer refers to a layer of coarse texture that covers a layer of fine texture, which is precisely due to the heterogeneity of the soil texture that causes the difference in the water infiltration process [21]. For example, when the homogeneous soil is loose sand soil and light loam soil and the layered soil is loose sand soil with light clay and light loam soil with light clay, through experiments with different irrigation volumes, it was found that due to the high water holding capacity of the light clay interlayer at a depth of 35–45 cm, its mass water content (18–28%) was far greater than the mass water content of the lower loose sand soil layer (6–9%) [22]. When the traditional upper soil and lower sand structure was used, the soil profile water content presented significant stratification, and the average profile moisture content of the Yellow River silt layer below 60 cm was 0.32 cm$^3$/cm$^3$, which was only 0.63 times that of the soil layer above 60 cm [19]. The sandwich structure led to a slow infiltration rate of water and prolonged the infiltration time, resulting in an increase in the upper soil moisture content of 22.52–29.33%. In this paper, the irrigation infiltration test was carried out by using the actual irrigation water quota in the field instead of fixed head infiltration. The results show that when the interlayer structure was the same and the irrigation amount was different, the interlayer had little influence on the surface soil moisture content, but had a great influence on the upper and lower soil moisture content of the interlayer. That is, when the irrigation amount was small, it was difficult for the water to penetrate through the interlayer to the lower layer, which is consistent with the research results of Tu et al. [14] and Dong et al. [23]. When fine soil was in the lower layer, the saturated water conductivity of soil was significantly lower than the effective saturated water conductivity of coarse soil in the lower layer and it was mainly determined by the water conductivity characteristics of fine soil [24,25]. In different irrigation methods, such as surface drip irrigation, subsurface drip irrigation, furrow irrigation, and border irrigation, among others, the different drip flow [26], drip irrigation volume [27], buried depth of drip irrigation belt [28], and furrow irrigation area will also affect the water movement pattern during soil infiltration.

To explore the actual soil water movement law closer to the cotton planting field in southern Xinjiang, the flooding irrigation method used in the soil column experiment in this paper is similar to the border irrigation method used in the field. Some studies have shown that under the border irrigation method, the soil water content after irrigation presents a layered phenomenon along the longitudinal direction; that is, the water content of the deep soil was the largest [29], and the longitudinal depth of the maximum water content increased with the increasing irrigation amount [30]. When the soil texture was consistent, the irrigation amount was 315 mm (T1), 270 mm (T2), and 225 mm (T3); in the soil moisture content at a longitudinal depth of 90 cm, the T1 treatment was 1.525% higher than the T2 treatment at the same time period after the irrigation stopped, while the T3 treatment had a small change within the 8 days after the irrigation stopped, and there was no obvious trend for increasing the water content like the T1 and T2 treatments. This indicates that the amount of irrigation water directly affected the vertical depth of water infiltration and the deep soil moisture content. When the interlayer soil changed, such as when the position (T5) and thickness (T6) of the interlayer changed, or when the irrigation amount was the same, the increase in the interlayer thickness had a greater impact on the soil water content than the increase in the interlayer position, which was also consistent with the research conclusions of Wang et al. [18] and others. In view of the blocking effect of the interlayer on the soil moisture content, this article used the method of Atkinson et al. [31] and Zhang et al. [32] to drill holes and irrigate sand. Through drilling holes, the interlayer soil in the heavily saline and alkaline soil was broken and then irrigated with sand to improve soil permeability and improve the soil environment. Its water infiltration effect (T4) was significantly better than that of the other treatments, improving the moisture content of the upper and lower layers of the interlayer soil.

*4.2. Effect of Interlayer Soil on Wetting Front and Cumulative Infiltration*

For research on the water movement characteristics of layered soil, indoor soil column or soil box infiltration and evaporation tests are mostly used, and most studies focus on the influence of sand embedment, soil layer ordering, and cover layer thickness on soil water characteristics [33–35]. Whether fine soil covers coarse soil or coarse soil covers fine soil, that is, when the upper and lower layers of soil are different, the overall infiltration rate will be reduced, and water infiltration will be hindered [36–38]. For example, Li et al. [39], through the layered soil stratification test of a soil box and the vertical one-dimensional infiltration test of layered soil conducted [40] in the laboratory, believed that the influence of interlayer soil on the infiltration process was manifested as permeability reduction, and the saturated water conductivity of the upper soil determined the average propulsion velocity in the lower soil. In addition, the infiltration of the interlayer soil is presented as a stable infiltration stage with a smaller infiltration rate and a smaller advance velocity for the wetting front. In this paper, the variation law of the infiltration rate of interlayer soil was consistent with the principle described above; that is, the studies were performed under different irrigation amounts, such as the T1, T2, and T3 treatments, but under the same interlayer texture condition—although the greater the irrigation amount, the greater the infiltration rate. The average infiltration rates were 0.0616 cm·h$^{-1}$, 0.0373 cm·h$^{-1}$, and 0.0261 cm·h$^{-1}$, respectively, while the infiltration rates of interlayer soil were 0.570 cm·h$^{-1}$, 0.426 cm·h$^{-1}$, and 0.420 cm·h$^{-1}$, respectively, indicating that stratified soil had the same order and different irrigation amounts, but was affected by the gravity of the irrigation amounts. The infiltration rate also decreased with the decrease in irrigation amount. For example, when the irrigation volume was 270 mm and 225 mm, the wetting front did not penetrate through the interlayer to reach the deeper soil. For different layers of soil thickness or location, the infiltration of soil water by the interlayer also showed significant differences. For example, compared with the T1 treatment, the infiltration rate of the interlayer soil was 0.126 cm·h$^{-1}$ and 0.078 cm·h$^{-1}$ after the T5 and T6 treatments, and were moved up by 10 cm and thickened by 10 cm. This indicates that the thickening of the interlayer increased the permeability inhibition effect of the soil compared with the upward movement of the interlayer, and the infiltration characteristics of different layered soil textures also showed differences, which is consistent with the research results of Li et al. [41]. Similarly, changes in the interlayer soil infiltration rate were bound to bring about changes in the wetting front and cumulative infiltration. For example, Li et al. [41] found that when the interlayer was 5–10, 10–15, and 20–15 cm away from the soil surface, the time for the wetting front to reach the interlayer interface, and the corresponding cumulative infiltration vary when there are different interlayer locations and soil layer rankings; in addition, when the interlayer level was higher, the time for the wetting front to reach the boundary line of the interlayer was shorter and the cumulative infiltration was smaller. In contrast, when the interbedded layer was deeper, the overall elevation of the infiltration rate curve led to a nonlinear increase in the cumulative infiltration and wetting front over time before water entered the interbedded boundary. However, the main reason to apply winter irrigation in southern Xinjiang is to facilitate salt leaching. Therefore, for the interlayer soil of heavy saline–alkali soil, the irrigation amount should be more than 315 mm.

In this paper, the analysis of the wetting front and cumulative infiltration of different interlayer treatments also showed a similar change rule. When the interlayer position was the same but the irrigation amount was different, the time for water to reach the interlayer interface shortened with the increased irrigation amount. The time for T1, T2, and T3 to reach the sandwich interface was 3474 min, 3957 min, and 4179 min, respectively. When the position and thickness of the interlayer changed, its wetting front and cumulative infiltration also changed. For example, after the T5 treatment moved upward and the T6 treatment thickened, the time for water infiltration to the interlayer interface was 4730 min and 7456 min, respectively, which is also consistent with the conclusions of Xu et al. [42], Li et al. [43], and Wu et al. [44], indicating that changes in the texture and location thickness

of the interlayer have a significant impact on soil water infiltration. Atkinson et al. [31], Liu et al. [45], Shi et al. [46] and Zhang et al. [47] found that drilling holes can loosen the soil and improve the soil environment, while sand filling can improve soil permeability by facilitating the formation of coarse particles. The results of the T4 treatment in this paper showed that compared with the other treatments, the soil moisture content in each soil layer was relatively stable, the stable infiltration rate of T4 was the highest (3.509 cm·h$^{-1}$), and the total time from the beginning to the end of infiltration was the shortest at only 495 min, which significantly improved the permeability reduction effect of the interlayer and promoted water infiltration. However, because the infiltration rate was too fast, it is necessary to formulate a reasonable irrigation quota to avoid deep leakage.

In summary, this study demonstrates that drilling and sand filling can break the water-blocking effect of the interlayers, and thus effectively facilitate soil infiltration and increase the moisture content of each soil layer. Although the interlayer structures simulated in this study cannot virtually represent the actual interlayers in the field, the changes in infiltration patterns before and after the drilling and sand filling potentially suggest a wide applicability in southern Xinjiang or other similar regions. As this paper mainly focused on the water infiltration variations under different irrigation amounts and different interlayer positions and depths, the influences of different soil texture [48], saturated water conductivity [49], and soil matric suction [50], as well as other characteristic parameters on water infiltration were not analyzed. Further studies are needed to simulate the water movement process of the infiltration rate, wetting front, and cumulative infiltration [12,51]. Meanwhile, for heavy saline soil, drilling and sand filling can be used to improve the soil condition. Other improvement measures or techniques, such as applying phosphogypsum [52], organic matter [53], biochar [54], humic acid [55], and other improvement products, can also help improve the soil and hydraulic properties of saline soil. Therefore, systematic research is also needed in this regard. In order to provide more reliable data and technical support for the development and utilization of interlayer soil in heavy saline–alkali land in southern Xinjiang.

## 5. Conclusions

By analyzing the soil water infiltration patterns under different treatments, the results show that when the irrigation amount was 315 mm, the wetting front was 87.8 cm, but when the irrigation amount was 225 mm, the wetting front was only 69.2 cm, which did not penetrate the interlayer soil. When the interlayer moved down to 50 cm away from the soil surface and 15 cm in thickness, the water infiltration rate decreased using 6000 min to complete the infiltration. The wetting front under the T5 and T6 was 71.4% and 77.3% of that under the T1, respectively; while it took only 3400 min for the drilling and sand filling treatment (T4) to penetrate the whole soil column (1 m), and the infiltration time was the shortest.

**Author Contributions:** B.W., J.Z. and Y.B. designed and supervised the research project. H.L. performed the experiments and collected the data. X.L. and B.Z. helped in data collection. H.L. and B.Z. analyzed the data and wrote the manuscript. H.L. and X.L. edited the manuscript. B.W. read and approved the final manuscript. All authors have read and agreed to the published version of the manuscript.

**Funding:** This work was supported by the Major Special Science and Technology Project of Xinjiang Province (2022A02007-3), National Key R&D Plan Projects (2021YFD1900805-04, 2022YFD190010404), National Natural Science Foundation of China (52269017), and the Xinjiang Tianshan Talent Leadership Training Project (2022TSYCLJ0069).

**Conflicts of Interest:** The authors declare no conflict of interest.

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
