# Peer review of "Influence of Interlayer Soil on the Water Infiltration Characteristics of Heavy Saline–Alkali Soil in Southern Xinjiang"

_agronomy, doi:10.3390/agronomy13071912_

Round 1
Reviewer 1 Report
Comments and Suggestions for Authors
In this manuscript the authors presented an interesting study on the influence of interlayer soil on the water infiltration. However, the authors do not respect the presentation the manuscript following the Authors Guide, and the paper does not present line numbers to report the comments and correction. In addition, the statistical analysis in the document is very poor, and the significance of different statistical test were not presented in the document. The manuscript can not be published in actual form, and the authors need to correct in this stage the following points :
· The authors need to add the line numbers in the document.
· The authors need to use international Unit system (first line in the introduction “hm2”)
· The authors need the rewrite this section “ Of the 1031×104 hm2 farmland suitable for agriculture in Xinjiang.. “
· The authors need to add the percentage of sand in soil profiles described in the section “Soil profiles in northwestern China are usually layered, and the common layered pro-files include sand on top with sand on bottom, sand on bottom with sand on top with sand on bottom, sand with sand on bottom with sand on bottom or sand with sand with a special interlayer.”
· In the section “The water holding capacity and infiltration of layered soil are very different from those of homogeneous soil” è See the reference “ DOI: 10.3390/su142013087 “ regarding the effect of salinity in water infiltration and soil hydraulic properties
· Remove “ The experimental soil was collected in December 2021 from the high-efficiency water-saving experimental demonstration base of Hailou Village, Hailou Town, Weigan River Irrigation District, Shaya County, Aksu Prefecture, Xinjiang.” this section is not relevant for the study.
· Correct “mu high-efficiency”
· Add the reference used for the determination of the bulk density, field capacity.
· The authors need to explain what they mean by “ Initial moisture content”
· The resolution of Figure 2 and Figure 5 needs to be improved.
Comments on the Quality of English Language
An extensive editing of English language is needed.
Author Response
1.The authors need to add the line numbers in the document.
Answer: line numbers have been added to the revised manuscript as required.
2.The authors need to use international Unit system (first line in the introduction “hm2”)
Answer: The units (e.g., hm2) have been changed to international metric unit “mm” or “ha”, depending on the context.
3.The authors need the rewrite this section “Of the 1031×104 hm2 farmland suitable for agriculture in Xinjiang.. “
Answer: L45-L49, Revised to “Among the 1031×104 ha of future usable wasteland in Xinjiang, the area of salinization is 515.11×104 ha, accounting for 49.93% of the total area. Among the total arable land in Xinjiang, the area containing interlayers is 32.5×104 ha, accounting for approximately 8.0% of the total arable land area.”
4.The authors need to add the percentage of sand in soil profiles described in the section “Soil profiles in northwestern China are usually layered, and the common layered pro-files include sand on top with sand on bottom, sand on bottom with sand on top with sand on bottom, sand with sand on bottom with sand on bottom or sand with sand with a special interlayer.”
Answer: Apologize for the confusion. TheL51-L53 has been modified to “In the field, layered soil profiles often include different interlayer patterns, such as sandy soil with a clay layer in the middle, or sandy soil with a sand layer in the middle, or sand and clay alternatively interlaid.”
5.In the section “The water holding capacity and infiltration of layered soil are very different from those of homogeneous soil” è See the reference “ DOI: 10.3390/su142013087 “ regarding the effect of salinity in water infiltration and soil hydraulic properties
Answer: L574, Cite this literature and the foreign literature related to the research content of this paper.
6.Remove “The experimental soil was collected in December 2021 from the high-efficiency water-saving experimental demonstration base of Hailou Village, Hailou Town, Weigan River Irrigation District, Shaya County, Aksu Prefecture, Xinjiang.” this section is not relevant for the study.
Answer: Removed as suggested.
7.Correct “mu high-efficiency”
Answer: revised into international metric unit “3333 ha”.
- Add the reference used for the determination of the bulk density, field capacity.
Answer: L92-L95 This sentence is revised to read: "After sampled with cutting rings, the soil bulk density was calculated from the dry soil mass per over the ring volume, and the field capacity is obtained by dividing the weight difference between the saturated and the air-dried soil over the air-dried soil mass.”
- The authors need to explain what they mean by “ Initial moisture content”
Answer: L95-L98 Sorry for the confusion. This part has been modified and now reads: “The initial moisture content is after sampling, in order to minimize the undesirable variations in soil moisture, all the soil was air dried to obtain comparatively consistent water content. Therefore, the soil moisture after air-drying was recorded as the initial soil moisture using TDR (Kaimeite, Zhuhai ,China)”.
10.The resolution of Figure 2 and Figure 5 needs to be improved.
Answer: Revised Figure 1 to Figure 5 with higher resolution are included in the revised manuscript.
Reviewer 2 Report
Comments and Suggestions for Authors
The study entitled “Influence of interlayer soil on the water infiltration characteristics of heavy saline-alkali soil in southern Xinjiang” is very interesting and the topic of the paper is valuable. However, the manuscript has some issues that need to be improved so that it can be elevated and appropriate for publication.
Some specific comments and suggestions for the authors are listed below:
1. More information should be given on the scope and the innovation of this study at the abstract.
2. Keywords should not include the same words like those into the title.
3. “Humidity” should be replaced with “soil moisture” and migration depth should be replaced with “wet head”, or “wet front”, or “soil moisture head”, or “soil moisture front”.
4. In Introduction, the second paragraph should be rewritten. Some sentences confuse the reader, e.g ”sand on top…interlayer” and “For example…water holding capacity[11]”. You are saying that hydraulic barrier is forming at both cases. Please explain better the meaning.
5. You should give the company names of the scientific tools described at the subsection 2.1. (mechanical analysis, TDR, etc)
6. Table 1 should be removed to the Results.
7. In Table 1 the soil texture designation is not in accordance with the bulk density and the field capacity which both refer to more light texture soils and not so cohesive as the silty loam. You should reconsider the designation of the soil texture. Also, the values of field capacity are not in accordance with the values of bulk density. When we have higher bulk density, what type of soil is indicated and what this means for FC?
8. Units of irrigation depths in abstract and in 2.2 subsection are not in accordance…please correct the mistakes. Also, irrigation depth is usually presented in mm. Maybe you should reconsider the units you use, in order to be in line with what it is commonly used worldwide.
9. At 2.3 subsection you should give more information on the experimental procedure. Did you keep a constant water supply, or did you keep a constant water load of 30 cm on the soil surface?
10. At subsection 2.3 you should give more information and maybe a figure of the soil columns, indicating the layers and their textures. It is completely confused for the reader to understand the structure of the layers.
11. In Results section you use the same titles in subsections with the Discussion section, while in results you make comments on some of the findings and figures. In my opinion, this befits into the Discussion section. You should not make comments about your findings in Results. You should only present the tables and figures. Besides, in Discussion you should analyze all your findings in comparison to other studies.
12. What do you mean by “improved infiltration”? What do you mean by “drilling sand irrigation”? These phrases confuse the reader. E.g. when we have coarse textured soils, “improved infiltration” may refer to achieving smaller infiltration rate (slower velocity of water), while in fine textured soils achieving “improved infiltration” may refer to bigger infiltration rate (faster velocity of water). Please define.
13. You are using the term “stable infiltration rate”, but you do not give information about “stable infiltration rate”. What does this value represent? Is it connected to a crucial hydraulic parameter?
14. About Figure 1. Your experiment lasted 2500 min, but you only took 6 to 8 experimental points of infiltration rate. Why didn’t you take more?
15. You mention only gravity force as the main force that is responsible for water motion. What about capillary forces at the beginning of the phenomenon? Also, in my opinion, since the water is moving into the unsaturated zone, you should also refer to Sorptivity, which plays fundamental role. Even just mentioning the above, will elevate your paper.
16. In Table 3 you give the linear regression parameters of cumulative infiltration (I) with wetting front (Zf), but you do not give the equations. Although you say it is linear, you should give the equations I=f(Zf).
17. In figure 5 we see the duration that water needs to reach each soil moisture sensor. What happened in 90 cm of T5 and T6 treatment? Why there is no change in soil moisture? One more thing: Why all the sensors of the same layer do not tend to the same soil moisture after time?
18. How did you simulate the Kostiakov equation with cumulative infiltration data? Did you use logarithmic scale? Simulation data along with experimental data should be shown in a figure. You should present the work that has been done.
19. Although the salinity and the pH of the soils are mentioned into the title of the manuscript, no information about these values are given into the text.
20. Maybe you should add the need for further investigation on more soil types, basins, etc. as a future prospect.
21. Most of the recent literature you use in your paper is from China, which is quite biased. I believe that you should use more worldwide references to strengthen your findings. Since the research is dealing with soil moisture at layered soils, maybe you could take under consideration the following article: Hydrodynamics of the Vadose Zone of a Layered Soil Column. Water 2023, 15, 221. https://doi.org/10.3390/w15020221
All in all, the topic of the paper is very interesting and valuable, but in order to be proper for publication and attract international readers, the spots mentioned above should be revised.
My suggestion is major revision.
Moderate editing of English language is required.
Author Response
1.More information should be given on the scope and the innovation of this study at the abstract.
Answer: Modify summary L14-L16. “Interlayer soil is common in southern Xinjiang, because interlayer can reduce the infiltration rate of soil water, according to the actual situation of interlayer soil in heavy saline-alkali cotton fields,” and L35 - L37. “The treatment measure of drilling first and then filling sand can be used as an effective improvement measure to increase the water infiltration rate of interlayer soil, and the operation is simple, and can be applied to the layered soil structure in the interlayer position of 60-80cm.”
- Keywords should not include the same words like those into the title.
Answer: L38 and L39, Change the keyword to: “southern xinjiang region; soil column test; drilling and sand filling; interlayer soil; water infiltration characteristics”
- 3. “Humidity” should be replaced with “soil moisture” and migration depth should be replaced with “wet head”, or “wet front”, or “soil moisture head”, or “soil moisture front”.
Answer: L30, Modify to “wet front of the soil moisture”
- In Introduction, the second paragraph should be rewritten. Some sentences confuse the reader, e.g ”sand on top…interlayer” and “For example…water holding capacity[11]”. You are saying that hydraulic barrier is forming at both cases. Please explain better the meaning.
Answer: L65-L68, Modify to: “When there is an interlayer in the soil profile, no matter clay over sand, or sand over clay, it will inhibit water infiltration [9], but if the soil is alternatively interlaid with clay and sand layers, the mixed structure will increase the overall water holding capacity of the profile [10,11].”
5.You should give the company names of the scientific tools described at the subsection 2.1. (mechanical analysis, TDR, etc)
Answer: L97-L98, Modify to: “Therefore, the soil moisture after air-drying was recorded as the initial soil moisture using TDR (Kaimeite, Zhuhai ,China).”
6.Table 1 should be removed to the Results.
Answer: L164 Table 1 moved to content 3.1
7.In Table 1 the soil texture designation is not in accordance with the bulk density and the field capacity which both refer to more light texture soils and not so cohesive as the silty loam. You should reconsider the designation of the soil texture. Also, the values of field capacity are not in accordance with the values of bulk density. When we have higher bulk density, what type of soil is indicated and what this means for FC?
Answer: L164, In accordance with the international classification standard for soil texture, the contents of silt and sand in Table 1 were revised and determined to be silty clay. The data of soil bulk density and field water capacity in Table 1 were all measured data after field sampling and were not modified.
8.Units of irrigation depths in abstract and in 2.2 subsection are not in accordance…please correct the mistakes. Also, irrigation depth is usually presented in mm. Maybe you should reconsider the units you use, in order to be in line with what it is commonly used worldwide.
Answer: The unit of irrigation volume in this paper was modified, and all m3·hm-2 was converted into mm.
- At 2.3 subsection you should give more information on the experimental procedure. Did you keep a constant water supply, or did you keep a constant water load of 30 cm on the soil surface?
Answer: L130-L132, increase “water of pre-designed volume (i.e., simulated irrigation height) was fed from the top of each column all at one time to start the infiltration test. The water heights of T1, T2 and T3 were 315 mm, 270 mm and 225 mm, respectively, and that of T4, T5 and T6 were all 315 mm.”
- At subsection 2.3 you should give more information and maybe a figure of the soil columns, indicating the layers and their textures. It is completely confused for the reader to understand the structure of the layers.
Answer: L140, In Figure 1, add a diagram of soil interlayer in soil column
Figure 1. Test soil column and sensor position and interlayer profile structure diagram
- In Results section you use the same titles in subsections with the Discussion section, while in results you make comments on some of the findings and figures. In my opinion, this befits into the Discussion section. You should not make comments about your findings in Results. You should only present the tables and figures. Besides, in Discussion you should analyze all your findings in comparison to other studies.
Answer: Following your suggestion, we have modified the Results by just presenting the observed patterns, and removed the comments into the Discussion.
- What do you mean by “improved infiltration”? What do you mean by “drilling sand irrigation”? These phrases confuse the reader. E.g. when we have coarse textured soils, “improved infiltration” may refer to achieving smaller infiltration rate (slower velocity of water), while in fine textured soils achieving “improved infiltration” may refer to bigger infiltration rate (faster velocity of water). Please define.
Answer: The expression of "drilling sand irrigation" is incorrect, and the full text is modified to "sand filling", and the full text of "Improved infiltration" is modified to "increase infiltration".
Layered soil profile structure
- You are using the term “stable infiltration rate”, but you do not give information about “stable infiltration rate”. What does this value represent? Is it connected to a crucial hydraulic parameter?
Answer: L190-L195, Add a description of the stable infiltration rate “The stable infiltration rate is the ratio of the cumulative infiltration amount after the stabilization of infiltration to the infiltration time. Since the interlayer divides the soil into "upper layer" and "lower layer", the water infiltration characteristics of different interlayer soils are different. As an important parameter of the water infiltration characteristics of interlayer soil, the stable infiltration rate reflects the infiltration characteristics of the overall structure of interlayer soil.”
- About Figure 1. Your experiment lasted 2500 min, but you only took 6 to 8 experimental points of infiltration rate. Why didn’t you take more?
Answer: L175 - L178, Add content “since the infiltration rate represents the change of infiltration depth with time, the 100cm soil layer is divided into 10 points according to the infiltration depth of each treatment in the soil column test. Therefore, the corresponding values of the depth of each treatment (experimental points) are between 7-10 points,”
- You mention only gravity force as the main force that is responsible for water motion. What about capillary forces at the beginning of the phenomenon? Also, in my opinion, since the water is moving into the unsaturated zone, you should also refer to Sorptivity, which plays fundamental role. Even just mentioning the above, will elevate your paper.
Answer: Modify L181-183, “combined with the adsorption of soil capillary force, soil has high soil water suction at the initial stage of infiltration and the corresponding infiltration rate is very high [18].”
- In Table 3 you give the linear regression parameters of cumulative infiltration (I) with wetting front (Zf), but you do not give the equations. Although you say it is linear, you should give the equations I=f(Zf).
Answer: L357 To better illustrate the linear relationships between the cumulative infiltration and the wetting fron migration distance, we list the fitting equations and R2 of different treatments in Table 3. It now reads:
|
Treatments |
Fitting equation(I= f(Zf)) |
R2 |
|
T1 |
I = 0.4603 Zf-2.7518 |
0.9963 |
|
T2 |
I = 0.4651 Zf-3.2838 |
0.9923 |
|
T3 |
I = 0.4243 Zf-2.3322 |
0.9971 |
|
T4 |
I = 0.8993 Zf+0.6224 |
0.9991 |
|
T5 |
I = 0.4118 Zf-2.3074 |
0.9921 |
|
T6 |
I = 0.4298 Zf-2.1399 |
0.9956 |
- In figure 5 we see the duration that water needs to reach each soil moisture sensor. What happened in 90 cm of T5 and T6 treatment? Why there is no change in soil moisture? One more thing: Why all the sensors of the same layer do not tend to the same soil moisture after time?
Answer: L383-L387 Add to “among them, the wetting front of T5 and T6 treatment at 25000 minutes after the start of the experiment was 81.0cm and 87.6cm, respectively. Therefore, at the depth of 90cm, the soil moisture content remained in a stable state of "no change" within 20 days after the start of irrigation.”
L412 - L418 Add to “due to the change of irrigation amount, thickness and position, the time of water infiltration to the next layer is changed. Therefore, the moisture content at the same soil depth at the same time is different, but it will still show a similar change rule after a certain period of time. For example, when the irrigation amount is different and the thickness of the interlayer is the same as that of the position (T1, T2 and T3),Or the position and thickness of the interlayer are different and the irrigation volume is the same (T5, T6). ”
- How did you simulate the Kostiakov equation with cumulative infiltration data? Did you use logarithmic scale? Simulation data along with experimental data should be shown in a figure. You should present the work that has been done.
Answer: L434 - L435, Add to “Therefore, the data is not logarithmically transformed into a linearized form of the equation [20] .”
- Although the salinity and the pH of the soils are mentioned into the title of the manuscript, no information about these values are given into the text.
Answer: We added the information of soil salinity and pH in Table 1. It now reads:
Table 1. Selected physical properties of the tested soil.
|
Soil depth (cm) |
Particle composition (%) |
Texture |
Bulk density (g·cm−3) |
Initial moisture content (cm3·cm−3) |
Field capacity (cm3·cm−3) |
Salt content (g·kg–1) |
pH |
||
|
<0.002 mm |
0.002–0.02 mm |
0.02-2 mm |
|||||||
|
0–10 |
5.87 |
65.50 |
28.63 |
Silty clay |
1.65 |
0.056 |
0.352 |
14.09 |
8.71 |
|
10–20 |
5.51 |
65.38 |
29.11 |
Silty clay |
1.56 |
0.065 |
0.337 |
11.68 |
8.70 |
|
20–30 |
6.16 |
69.40 |
24.44 |
Silty clay |
1.51 |
0.073 |
0.330 |
10.79 |
8.70 |
|
30–40 |
5.88 |
68.07 |
26.05 |
Silty clay |
1.61 |
0.071 |
0.400 |
10.86 |
8.69 |
|
40–50 |
4.32 |
65.82 |
29.86 |
Silty clay |
1.38 |
0.063 |
0.329 |
21.18 |
8.74 |
|
50–60 |
5.77 |
74.03 |
20.20 |
Silty clay |
1.56 |
0.071 |
0.312 |
24.07 |
8.67 |
|
60–70 |
6.75 |
72.65 |
20.60 |
Silty clay |
1.73 |
0.069 |
0.284 |
29.97 |
8.79 |
|
70–80 |
4.68 |
62.12 |
33.20 |
Silty clay |
1.69 |
0.067 |
0.398 |
24.06 |
8.78 |
|
80–90 |
4.29 |
58.16 |
37.55 |
Silty clay |
1.55 |
0.063 |
0.410 |
19.77 |
8.77 |
|
90–100 |
5.41 |
71.00 |
23.59 |
Silty clay |
1.57 |
0.064 |
0.381 |
16.12 |
8.81 |
- Maybe you should add the need for further investigation on more soil types, basins, etc. as a future prospect.
Answer: L561-578, Modify discussion section, “In summary, this study demonstrates that drilling and sand filling can break the water-blocking effect of the interlayers, and thus effectively facilitate soil infiltration and increase the moisture content of each soil layer. Although the interlayer structures simulated in this study cannot virtually represent the actual interlayers in the field, the changes of infiltration patterns before-after the drilling and sand filling potentially suggest a wide applicability in southern Xinjiang or other similar regions。Since this paper only analyzed the rule of water infiltration under the change of irrigation amount and the position and depth of interlayer, the influence of different soil texture [48], saturated water conductivity [49] and soil matric suction [50] and other characteristic parameters on water infiltration was analyzed.Further studies are needed to simulate the water movement process of infiltration rate, wetting front and cumulative infiltration [12,51].Meanwhile, for heavy saline soil, drilling and sand filling are used to improve the soil condition. Other improvement measures or techniques, such as applying phosphogypsum[52], organic matter [53], biochar [54], humic acid [55] and other improvement products, can improve the soil and hydraulic properties of saline soil. Systematic research is also needed in this regard.In order to provide more reliable data and technical support for the development and utilization of interlayer soil in heavy saline-alkali land in southern Xinjiang.”
- Most of the recent literature you use in your paper is from China, which is quite biased. I believe that you should use more worldwide references to strengthen your findings. Since the research is dealing with soil moisture at layered soils, maybe you could take under consideration the following article: Hydrodynamics of the Vadose Zone of a Layered Soil Column. Water 2023, 15, 221. https://doi.org/10.3390/w15020221
Answer: L569, Cite this literature and the foreign literature related to the research content of this paper.
Round 2
Reviewer 2 Report
Comments and Suggestions for Authors
I have read carefully the revised manuscript and I believe that the authors made some serious efforts and the paper has now been elevated as most of the dark spots have been cleared up. However, authors must revise Table 1, as the characterization of the soils and the values of bulk density are not in accordance. The values of bulk density are too high for such cohesive soil as the silty clay. The values of bulk density refer to coarse soils. Hence, Table 1 must be revised. I also believe that the authors should checkout throughout the text for spelling and syntax errors.
Comments on the Quality of English LanguageI believe that the manuscript still needs a minor revision in English language. You should check out all the text, line by line, to correct any spelling or syntax mistakes, or phrases that may confuse the reader.
Author Response
- As the characterization of the soils and the values of bulk density are not in accordance. The values of bulk density are too high for such cohesive soil as the silty clay. The values of bulk density refer to coarse soils.
Answer: We understand your concerns that the soil bulk density seems a bit too high for coarse soils. However, soil bulk density is not only related to texture, but also closely associated with salt content, where higher salt content often results in heavier bulk density. Since the soil investigated in this study was heavy saline-alkali soil, the bulk density recorded in the field also reflected the impacts of soil salt content on bulk density.
- You should check out all the text, line by line, to correct any spelling or syntax mistakes, or phrases that may confuse the reader.
Answer: The original manuscript has been edited by Language Editing Service, and the revised text has been proofread by an experienced researcher.
